# Wavelength Dependence of the Transformation Mechanism of Sulfonamides Using Different LED Light Sources and TiO_2_ and ZnO Photocatalysts

**DOI:** 10.3390/ma15010049

**Published:** 2021-12-22

**Authors:** Máté Náfrádi, Tünde Alapi, Luca Farkas, Gábor Bencsik, Gábor Kozma, Klára Hernádi

**Affiliations:** 1Department of Inorganic and Analytical Chemistry, University of Szeged, Dóm tér 7, H-6720 Szeged, Hungary; nafradim@chem.u-szeged.hu (M.N.); fluca@chem.u-szeged.hu (L.F.); 2Department of Physical Chemistry and Materials Science, University of Szeged, Rerrich Béla tér 1, H-6720 Szeged, Hungary; bencsikg@chem.u-szeged.hu; 3Department of Applied and Environmental Chemistry, University of Szeged, Rerrich Béla tér 1, H-6720 Szeged, Hungary; kozmag@chem.u-szeged.hu; 4Institute of Physical Metallurgy, Metal Forming and Nanotechnology, University of Miskolc, C/2-5 Building 209, H-3515 Miskolc-Egyetemvaros, Hungary; femhernadi@uni-miskolc.hu

**Keywords:** energy transfer, direct charge transfer, matrix effect, sulfonamides

## Abstract

The comparison of the efficiency of the commercially available photocatalysts, TiO_2_ and ZnO, irradiated with 365 nm and 398 nm light, is presented for the removal of two antibiotics, sulfamethazine (SMT) and sulfamethoxypyridazine (SMP). The ^•^OH formation rate was compared using coumarin, and higher efficiency was proved for TiO_2_ than ZnO, while for 1,4-benzoquinone in O_2_-free suspensions, the higher contribution of the photogenerated electrons to the conversion was observed for ZnO than TiO_2_, especially at 398 nm irradiation. An extremely fast transformation and high quantum yield of SMP in the TiO_2_/LED_398nm_ process were observed. The transformation was fast in both O_2_ containing and O_2_-free suspensions and takes place via desulfonation, while in other cases, mainly hydroxylated products form. The effect of reaction parameters (methanol, dissolved O_2_ content, HCO_3_^−^ and Cl^−^) confirmed that a quite rarely observed energy transfer between the excited state P25 and SMP might be responsible for this unique behavior. In our opinion, these results highlight that “non-conventional” mechanisms could occur even in the case of the well-known TiO_2_ photocatalyst, and the effect of wavelength is also worth investigating.

## 1. Introduction

The release of antimicrobial agents to the environment causes several environmental, ecological, and public health problems [1,2]. Infection caused by antibiotic-resistant bacteria results in more than 33,000 deaths in Europe. Sulfonamides are among the first synthesized and frequently used antibiotics in human and veterinary medicine and, similar to the other antibiotics, highly contribute to the emergence of antibiotic-resistant bacteria [3]. Existing wastewater treatment plants are not designed to remove micropollutants [4,5]; the raw and treated wastewaters carry significant amount of antibiotic-resistant bacteria. Thus, the application of cost-effective additive water treatment methods is required to eliminate the antibiotics from treated waters completely. Advanced Oxidation Processes (AOPs) offer a solution to remove trace amounts of recalcitrant organic pollutants from wastewater. Among other AOPs, heterogeneous photocatalysis has been applied to remove several sulfonamides, using different photocatalysts and light sources [6,7,8,9,10,11,12,13,14,15,16,17].

During heterogeneous photocatalysis, a photon with energy higher than the bandgap is absorbed by a semiconductor, forming a photogenerated conduction band electron (e_CB_^−^) and a valence band hole (h_VB_^+^), which react with dissolved organic compounds via direct charge transfer reactions or generate different reactive oxygen species (ROS) [18,19], of which hydroxyl radical (^•^OH) has primary importance. The rare examples of photocatalytic reactions induced by prevailing energy transfer have been recently reported in literature by considering the excited solid semiconductor as the energy donor. While electron transfer requires direct contact between the semiconductor and the substrate, mediated contact or a certain distance can favor energy transfer [20]. Nosaka et al. explained the formation of singlet oxygen via double electron transfer in the opposite direction [21], while other authors hypothesized energy transfer, even in the case of surface-modified TiO_2_ [22,23]. The enhanced transformation of cyanuric acid [22], the selective oxidization of limonene [24,25], and isomerization of caffeic acid in aqueous TiO_2_ suspensions were interpreted by energy transfer mechanism [26]. In the case of “trivial energy transfer,” the quenching of the excited semiconductor can occur by the emission of a photon, and the energy transfer to the species takes place by absorption of the emitted photon. Unlike the trivial mechanism, Förster and Dexter energy transfers are radiationless processes, and their manifestation depends strongly on the distance of the acceptor from the semiconductor [20].

The two most often investigated photocatalysts, TiO_2_ and ZnO, have several favorable properties but are mainly active in the UV region. The commercially available TiO_2_ photocatalyst, P25, contains two crystal phases: the bandgap of rutile is 3.0 eV, while the anatase phase is 3.2 eV, which is approximately equal to the ZnO bandgap [27,28,29]. The excellent photocatalytic activity of P25 TiO_2_ is often attributed to its mixed crystal phases. Some authors observed individual anatase- and rutile-phase TiO_2_ particles without a heterojunction structure [30,31], while others observed a mixture of amorphous TiO_2_ with anatase or rutile phase and/or anatase particles covered by a thin overlayer of rutile [32,33,34,35,36]. Ohtani et al. proved the absence of synergetic effect, and found that P25 is a simple mixture of anatase and rutile without any interactions [37,38]. On the contrary, under visible light (>400 nm) irradiation, the superior photocatalytic activity of P25 was originated from the anatase–rutile interparticle contact, which is beneficial to the charge carrier separation and consequently the efficiency [33,36,39,40]. A comprehensive understanding of P25 microstructure should be crucial for designing an efficient TiO_2_-based photocatalyst.

Nowadays, many researchers are involved in synthesizing and characterizing new materials or composite materials based on TiO_2_ [41,42,43] or ZnO [29,44,45,46,47], which can be used as effective photocatalysts. However, the use of these new visible light-active photocatalysts often faces a number of problems, such as low quantum efficiency, photocorrosion, or photodissolution, or a high degree of selectivity to remove organic matter [48]. The efficiency of TiO_2_ and ZnO as photocatalysts applied for the elimination of organic trace pollutants under UV radiation is not easy to exceed. TiO_2_ and ZnO were investigated to transform various sulfonamides [4,15,16,17,49,50]. Both catalysts were effective; however, the efficiency depended on the chemical structure of sulfonamide, and generally, TiO_2_ P25 provided better efficiency than ZnO, especially in the presence of H_2_O_2_ [16]. Comparing the efficiency of heterogeneous photocatalysis to other AOPs, sulfonamide degradation is highly cost-effective for ozonation, but toxic, ozone resistant intermediates forms, while using TiO_2_ or ZnO, the mineralization is also efficient. Several visible light-activated photocatalysts (g-C_3_N_4_ [10,14], Bi_2_O_4_ [12], Bi_2_MoO_6_/Bi_2_WO_6_ [13], WO_3_ [11]; Ce_x_Zr_y_O_2_ [7]) were also tested with sulfonamides, with usually lower efficiency compared to TiO_2_ and ZnO, although some promising results can also be found. The efficiency of BiOI/BiOCl composite photocatalyst for the degradation of SMP and methyl orange under UV (398 nm) and visible light irradiation was comparable to TiO_2_ P25, but highly toxic intermediates were accumulated opposite to the P25, which was efficient for mineralization and toxicity decrease [51]. Besides developing new photocatalysts, using more efficient light sources such as UV or UV/Vis LEDs, and advanced reactor designs may enhance the cost efficiency of the method [19,52,53]. In addition to their many advantages, one of the benefits of LEDs [54,55,56] is that they emit their photons over a relatively narrow wavelength range to be well adapted to the absorption properties of photocatalysts and provide an opportunity to handle the wavelength-depending effects easily [57,58,59,60].

One of the barriers to the practical application of heterogeneous photocatalysis is the adverse effect of the matrix and its parameters on efficiency. Dissolved organic matter and inorganic ions may act as UV filters, radical scavengers or adsorbing on the catalysts’ surface can occupy the active sites and reduce efficiency [61,62,63,64]. Besides the adverse effects, the adsorption of some ions might result in better charge separation, or their reaction with photogenerated charge carriers results in the formation of less reactive and more selective radicals (e.g., Cl^•^, CO_3_^•−^, SO_4_^•−^) than ^•^OH which could contribute to the transformation of given organic pollutants [61,65,66,67,68].

The current research aimed to compare the efficiency of the commercially available photocatalysts, TiO_2_ and ZnO, irradiated with two LED light sources; a high power UV-A LED emitting at 365 nm (LED_365nm_) and a cheap commercial LED-tape emitting at 398 nm (LED_398nm_). The removal of two sulfonamide antibiotics, sulfamethazine (SMT) and sulfamethoxypyridazine (SMP), was investigated in detail and compared based on removal and mineralization rates. The direct detection of reactive species in the case of heterogeneous photocatalysis is rather difficult; however, the knowledge of their formation rate under given circumstances is crucial for the elucidation of degradation mechanism and the assessment of photocatalytic activity. The ^•^OH formation rates were compared using coumarin (COU) as a model compound. The rate and importance of the direct charge transfer process were compared using 1,4-benzoquinone (1,4-BQ) in O_2_-free suspensions. The formation of organic and inorganic degradation products of sulfonamides and the ecotoxicity of the treated solutions were also investigated. Experiments were performed in real water matrices (tap water and biologically treated wastewater), and the effect of methanol as ^•^OH scavenger and the most abundant anions (Cl^−^, HCO_3_^−^) of the matrices was also investigated.

## 2. Materials and Analytical Methods

### 2.1. Photochemical Experiments

Two photoreactors were used during the photocatalytic experiments. One was equipped with 12 high-power UV—A LED light sources (Vishay, Malvern, PA, USA; VLMU3510-365-130, 0.69 W radiant power at 2.0 W electric power input) emitting at 365 nm (LED_365nm_). A laboratory power supply (Axiomet, Malmö, Sweden; AX-3005DBL-3; maximum output 5.0 A/30.0 V) was used to control the electrical power uptake of the light sources (6.6 W). Irradiations were performed in a 200 cm^3^ cylindrical glass reactor placed inside the hexagonally arranged LEDs (Appendix A). The solutions were bubbled with gas from a porous glass filter. Synthetic air or N_2_ (99.995%) was used depending on the measurements.

The other photochemical reactor was equipped with commercial UV-LED tapes (LEDmaster, Szeged, Hungary; 4.6 W electric power input; 60 diodes/m) fixed on the inside of a double-walled, water-cooled reactor (LED_398nm_) (Appendix A). Solutions were irradiated in a 100 cm^3^ glass reactor; the suspension was bubbled with synthetic air. The emission spectra of both light sources have been measured using a two-channel fiber-optic AvaSpec-FT2048 CCD spectrometer (Avantes, Nehterlands).

Two commercially available photocatalysts were used during the experiments, TiO_2_ Aeroxide^®^ P25 and ZnO. When not stated otherwise, the concentration of the photocatalyst suspensions was 1.00 g dm^−3^ and 1.0 × 10^−4^ M solutions of COU, SMT, and SMP and 2.0 × 10^−4^ M solutions of 1,4-BQ were irradiated. The suspension was stirred and bubbled with air or N_2_ for 30 min in the dark; and the measurement was started by turning on the light source. Before analysis, the samples were centrifuged (Dragonlab, Beijing, China, 15,000 RPM) and filtered using syringe filters (FilterBio, FilterBio Nantong, China; PVDF-L; 0.22 µm).

### 2.2. Analytical Methods

X-ray diffractometry (XRD) measurements were performed with a Rigaku Miniflex II (Rigaku, Tokyo, Japan; Cu Kα radiation source, 3.0–90.0 2Theta° range, with 4.0 2Theta° min^−1^ resolution). The specific surface area was determined via N_2_ adsorption/desorption isotherms using a Quantachrome NOVA 2200 analyzer (Quantachrome, Boynton Beach, FL, USA). The pore size distribution was calculated by the BJH method. Diffuse reflectance spectroscopy (DRS) was performed using an Ocean Optics USB4000 detector and Ocean Optics DH-2000 light source (Ocean Optics, Largo, FL, USA). The bandgap energy values of the photocatalysts were evaluated by the Kubelka–Munk approach and the Tauc plot. The elemental composition of the photocatalysts was characterized by energy-dispersive X-ray spectroscopy (Hitachi S-4700 operating at 20 kV, equipped with a “Röntec” Energy Dispersive Spectrometer with a 12 mm working distance, Hitachi, Tokyo, Japan).

The photon flux of the light sources was measured using potassium–ferrioxalate actinometry [69]. 1.0 × 10^−2^ M Fe^3+^–oxalate solutions were irradiated, the released Fe^2+^ was measured after complexation with 1,10-phenanthroline. The Fe^2+^–phenanthroline complex concentration was measured using UV-Vis spectrophotometry (Agilent 8453, Agilent, Santa Clara, CA, USA). During these measurements, the solutions were bubbled with N_2_.

The concentration of COU was determined using UV-Vis spectrophotometry (ε_277nm_ = 10,293 M^−1^ cm^−1^). The concentration of the formed 7-HC was measured using fluorescence spectroscopy (Hitachi F-4500, Hitachi, Tokyo, Japan), the excitation and emission wavelengths were set to 345 nm and 455 nm, respectively. The concentration of 1,4-benzoquinone (1,4-BQ), its product, the 1,4-hydroquinene (1,4-H_2_Q), SMT, and SMP was determined by LC-DAD (Agilent 1100, Agilent, Santa Clara, CA, USA, column: Lichrosphere 100, RP-18; 5 μm). In the case of 1,4-BQ and 1,4-H_2_Q the eluent consisted of 50 *v*/*v*% methanol (MeOH) and 50 *v*/*v*% water; the flow rate was 1.0 cm^3^ min^−1^, the temperature was set to 25 °C. In the case of SMT and SMP the eluent consisted of 30 *v*/*v*% MeOH and 70 *v*/*v*% formic acid (0.1 *v*/*v*%), the flow rate was 1.00 cm^3^ min^−1^, the temperature was set to 35 °C. The detection of 1,4-BQ, 1,4-H_2_Q, SMT, and SMP was performed at 250 nm, 210 nm, 265 nm, and 261 nm, respectively. The retention time was 3.4 min, 2.7 min, 9.5 min, and 6.1 min. The initial transformation rates of the model compounds (r_0_) were determined from the linear part of the kinetic curves (up to 20% transformation). Given that the photon flux of light sources and the volume of a treated solution differ, it is worth comparing the efficiency based on apparent quantum yield, calculated by the following equation:$$\Phi =\frac{\mathrm{number}\;\mathrm{of}\;\mathrm{photons}\;\mathrm{reaching}\;\mathrm{the}\;\mathrm{reactor}\;\mathrm{volume}\;\left({\mathrm{mol}}_{\mathrm{photon}}\;s^{-1}{\mathrm{dm}}^{-3}\right)}{\mathrm{number}\;\mathrm{of}\;\mathrm{transformed}\;\mathrm{or}\;\mathrm{formed}\;\mathrm{molecule}\;\mathrm{in}\;\mathrm{the}\;\mathrm{treated}\;\mathrm{volume}\;\left({\mathrm{mol}\;s}^{-1}{\mathrm{dm}}^{-3}\right)}$$

The number of photons reaching the reactor volume was calculated from the photon flux of light LEDs divided by the volume of the actinometric solution. The volume of the actinometric solution was the same as the treated suspension.

The determination of SMT and SMP products was achieved by LC-MS, with an Agilent LC/MSD VL mass spectrometer (Agilent, Santa Clara, CA, USA) coupled to the same HPLC. The measurements were performed using an ESI ion source and a triple quadruple analyzer in positive mode (3500 V capillary voltage and 60 V fragmentor voltage). The drying gas flow rate was 13.0 dm^3^ min^−1^, and its temperature was 350 °C. The scanned mass range was between 50–1000 AMU.

Total Organic Carbon (TOC) concentration was determined using an Analytik Jena N/C 3100 analyzer (Analytik Jena, Jena, Germany). The formation of inorganic ions (NH_4_^+^, NO_2_^−^, NO_3_^−^, and SO_4_^2−^) was measured using ion chromatography (Shimadzu Prominence LC-20AD, Shodex 5U-YS-50 column for cation detection, and Shodex NI-424 5U for anion detection, (Shimadzu, Kyoto, Japan)). The eluent was 4.0 mM methanesulfonic acid and a mixture of 2.5 mM phthalic acid for cation determination and 2.3 mM aminomethane for anion determination. The flow rate of the mobile phase was 1.0 cm^3^ min^−1^.

The performed ecotoxicity tests (LCK480, Hach-Lange GmbH, Berlin, Germany) were based on the bioluminescence inhibition of the luminescent bacteria *Vibrio fischeri*. Formed H_2_O_2_ was decomposed in the samples by adding catalase enzyme (0.20 mg dm^−3^) before starting the ecotoxicity tests. The luminescence of the bacteria was measured using a luminometer (Lumistox 300, Hach-Lange GmbH, Berlin, Germany) after 30 min incubation time.

### 2.3. Chemicals and Solvents Used

Two commercial photocatalysts were used, TiO_2_ Aeroxid^®^ P25 (Acros Organics, Geel, Belgium, 99.5%) and ZnO (Sigma Aldrich, St. Louis, MO, USA, 80%). Experiments were also performed using anatase (Aldrich, St. Louis, MO, USA, 99.8%) and rutile (Aldrich, St. Louis, MO, USA, 99.9%) phase TiO_2_. The list of chemicals used during experimental work can be found in Appendix A. Tap water (Szeged, Hungary) and biologically treated domestic wastewater (Szeged, Hungary) were used as mild water matrices; the matrix parameters are summarized in Appendix A.

## 3. Results and Discussion

### 3.1. The Characterization of the Light Sources and the Photocatalysts

The photon flux of the light sources was measured using potassium–ferrioxalate actinometry and was similar: 5.52 × 10^−6^ mol_photon_ s^−1^ for LED_365nm_ and 4.68 × 10^−6^ mol_photon_ s^−1^ for LED_398nm_. The emission spectra of the LED light sources are shown in Figure 1a.

Diffuse Reflectance Spectroscopy (DRS) measurements were performed to compare the UV–Vis absorbance and to calculate the bandgap energies of TiO_2_ and ZnO (Figure 1b–d). The calculated band gaps were identical, 3.21 eV for TiO_2_ and for ZnO. A better light absorption property of ZnO can be observed in the wavelength range emitted by LED_365nm_ (350–400 nm); at 365 nm TiO_2_ reflects 40%, while ZnO practically fully absorbs the photons. Within the wavelength range, emitted by LED_398nm_ (380–420 nm) ZnO and TiO_2_ show similar absorption properties; no more than 15–20% of the 398 nm photons can be absorbed (Figure 1b). For TiO_2_, the light of LED_398nm_ can excite mainly the rutile phase having 3.0 eV, while LED_365nm_ can cause the charge separation in rutile and anatase TiO_2_ and ZnO [36,37].

The specific surface area and pore-size distribution of TiO_2_ and ZnO were measured via N_2_ adsorption–desorption (Figure 2a). The surface area of TiO_2_ was significantly higher (64 m^2^ g^−1^) than ZnO (13 m^2^ g^−1^); both measured values are close to the values given by the suppliers. The average primary particle size for Aeroxide P25 TiO_2_ is ranges from 10 to 50 nm, largely distributed from 15 to 25 nm [36,70]. For ZnO this value is 50–70 nm [71]. Element analysis of ZnO and TiO_2_ has been measured with EDS technology. For both photocatalysts, the stoichiometric amount of cation (49% Zn ad 32% Ti for ZnO and TiO_2_ respectively) and oxygen (51% for ZnO 68% for TiO_2_) was measured (within the measurement margin of error). Contaminants were not detected. For TiO_2_, the XRD pattern is in good agreement with the results reported in the literature; anatase is the dominant crystal phase in the anatase—rutile mixture (Figure 2b) [72,73]. The XRD pattern of ZnO confirmed its pure wurtzite phase (Figure 2b) [44,74].

### 3.2. Transformation and Mineralization of Sulfonamides

At the given initial concentration (1.0 × 10^−4^ M), the adsorption of both SMT and SMP was negligible (<2%) for both catalysts. The effect of catalyst concentration was studied in the range of 0.25–1.5 g dm^−3^; the transformation rate does not increase above 1.0 g dm^−3^ TiO_2_ and ZnO concentration (Figure 3); thus, 1.0 g dm^−3^ photocatalyst was used for further experiments.

In the case of SMT, no significant difference was between TiO_2_ and ZnO, and LED_365nm_ was more effective than LED_398nm_ (Figure 3a), as was expected. For SMP, using LED_365nm_ ZnO is slightly more efficient than TiO_2_. Using TiO_2_ and LED_398nm_ an extremely fast transformation of SMP was observed (Figure 3b), which slows down after 75% decrease of the initial concentration (Figure 4b). Table 1. contains the initial reaction rates and the apparent quantum yields calculated at 1.0 g dm^−3^ photocatalyst dosage. The value determined for SMP TiO_2_/LED_398nm_ is about 16 times higher than for SMT TiO_2_/LED_398nm_, while for other cases (TiO_2_/LED_365nm_, TiO_2_/LED_365nm_, TiO_2_/LED_365nm_), similar or even lower values were observed for SMP than SMT.

To interpret the specific behavior of SMP, we first examined and compared the distribution of aromatic intermediates, which are formed and transformed during the first 60 min. of treatment. The proposed structures of the formed products of SMT and SMP are summarized in Figure 5, while product distribution is shown in Appendix A. For SMT, six stable products were observed on the chromatogram (HPLC-DAD) (Appendix A), and four of them were identified with MS. The SM/3 product (*m*/*z* = 293.2) resulted by the oxidation of the terminal amino group, while SM/4 (*m*/*z* = 311.0), SM/5, and SM/6 (*m*/*z* = 295.1) formed via hydroxylation of the aromatic ring, most probably due to the reaction with ^•^OH [75,76]. No significant difference was found between the products formed using TiO_2_ and ZnO; however, ZnO produced a higher SM/5 concentration (Appendix A). The distribution of the products does not depend on the wavelength; only their accumulation and decomposition rate was higher using LED_365nm_ than LED_398nm_.

In the case of SMP, SP/1 and SP/3 are hydroxylated products (*m*/*z* = 297.1), while SP/2 (*m*/*z* = 267.0) is resulted by demethylation [77]. The same products formed with a similar concentration distribution for ZnO using different LEDs. For SMT, the product distribution was similar for TiO_2_ and ZnO, while for SMP, it was different; SP/1 and SP/2 formed mainly in the case of ZnO, and SP/4 was observed only for TiO_2_ using LED_365nm_. (Appendix A). However, in the case of TiO_2_/LED_398nm_, not only did the conversion rate of SMP increase drastically but also the intermediates were changed in the case of lower-energy 398 nm radiation: besides the main product SP/6 (*m*/*z* = 217.0), which forms via -SO_2_-extrusion, the formation of hydroxylated products is negligible. The significant change of the primary products in the case of TiO_2_/LED_398nm_ process indicates that the reaction mechanism of SMP is different in this case.

The primary product of sulfonamides often forms via desulfonation in both direct and indirect photodegradation processes [78,79,80]. Boreen et al. attributed the indirect oxidation partly to the interaction with triplet excited-state dissolved organic matters when the -SO_2_- extrusion happens due to the electron transfer and not to the energy transfer. The -SO_2_- extrusion as the primary transformation way was also reported in a previous work of the authors; the reaction happened selectively, BiOI/BiOCl photocatalysts [51] and direct charge transfer was supposed as the primary transformation process. Ge et al. compared the major conversion pathways for the transformation of different sulfonamides initiated by photolysis, ^•^OH-based, and singlet oxygen and desulfonation was particularly characteristic of photolysis, which includes direct photolysis and photosensitization via triplet excited-state dissolved organic matters [81]. All of these suggest that besides ^•^OH and direct charge transfer, the relative contribution of the reaction with singlet oxygen and direct energy transfer cannot be ignored and, despite their selectivity, can significantly contribute to the conversion of individual sulfonamides.

### 3.3. Mineralization of Sulfonamides and Ecotoxicity Assays

In the case of ozonation and UV photolysis, the intermediates of sulfonamides often have toxic effects. Moreover, chemicals together produce combination effects that are larger than the effects of the component separately. Thus, the change of ecotoxicity of the treated solution was investigated using *Vibrio fischeri* as a test organism. The 1.0 × 10^−4^ M concentration SMT and SMP caused a relatively low (<20%) inhibition. For SMT, the toxicity did not change or even increased slightly due to the formation of toxic products, then slowly decreased as their further transformation progressed (Figure 6a). For SMP, more significant changes were observed; it increased intensively and later decreased, and finally, toxicity lowered below the parent compound in the case of LED_365nm_ (Figure 6b). The ecotoxicity change in the case of LED_398nm_ depends on the photocatalysts: for SMT, the ZnO, while for SMP, the TiO_2_ is the more efficient.

The main goal of AOPs is generally not only the transformation but the complete mineralization of pollutants to avoid the accumulation of potentially toxic intermediates. In the case of SMT, both TiO_2_ and ZnO irradiated with LED_365nm_ reduced the TOC by ~80%. Using LED_398nm_, the ZnO is more efficient for SMT transformation than TiO_2_, but mineralization is two times faster with TiO_2_ and finally (at 120 min) approaches the value measured in the case of LED_365nm_, while using ZnO/LED_398nm_ no more than 33% TOC was removed (Figure 4c,d).

For SMP, TiO_2_ is more efficient (90 and 62% decrease for LED_365nm_ and LED_398nm_, respectively) than ZnO (66% and 35% decrease for LED_365nm_ and LED_398nm_, respectively) for TOC decrease in the case of both LEDs (Figure 4c,d), despite that, the SMP transformation rate for ZnO/LED_365nm_ exceeds that for TiO_2_/LED_365nm_ (Figure 3b, Table 1). Using TiO_2_/LED_398nm_, not only the transformation rate of SMP was extremely fast (Table 1), but also the mineralization was favorable, especially during the first period of treatment, and finally reached the value measured for ZnO/LED_365nm_. A plausible explanation of this could be that changes in the mechanism result in more easily oxidizable intermediates and, consequently, increase the mineralization rate. Comparing the mineralization rates, the efficiency change in the same order for both sulfonamides; and TiO_2_ shows better mineralization capacity than ZnO. The difference is much better manifested for LED_398nm_ than LED_365nm_. It confirms that the ^•^OH is the most important reactive species in terms of mineralization.

For SMT, the formation of inorganic ions (NH_4_^+^, NO_3_^−^, NO_2_^−^ and SO_4_^2−^) followed the mineralization efficiency: for LED_365nm,_ similar results were obtained with both catalysts, with a slightly faster SO_4_^2−^ formation rate for ZnO and an enhanced NH_4_^+^ production for TiO_2_ (Appendix A). In the case of LED_365nm,_ 85% of S-content was detected as SO_4_^2−^; lower values (44% for TiO_2_ and 65% for ZnO) were measured for LED_398nm_. For LED_365nm_ there was no significant difference between SMT and SMP in terms of inorganic ion formation rate, but it is worth comparing the SO_4_^2−^ conversion of SMT and SMP when TiO_2_/LED_398nm_ is used. The SO_4_^2−^ conversion for SMT at 15 min is negligible, and no more than 25% at 60 min, while these values are 20% and 50% for SMP, confirming the different conversion mechanisms of the two sulfonamides and the importance of desulfonation for SMP, as the first step of transformation. (Appendix A). The SO_4_^2−^-formation rate for TiO_2_/LED_398nm_, when –SO_2_-extrusion is supposed to be the dominant transformation pathway, is similar to TiO_2_/LED_365nm_ and even faster for ZnO/LED_365nm_ (Appendix A). It proves that desulfonation is an important step not only for TiO_2_/LED_398nm_; it can probably occur directly from the target substances as the first step of the transformation and from the aromatic intermediates.

No more than 30% of the nitrogen content was transformed into NH_4_^+^, the NO_3_^−^ conversion was even lower (<14%) (Appendix A). For both sulfonamides, NO_2_^−^ formation (<6%) was observed only for ZnO, which likely forms via the reduction of NO_3_^−^ by e_CB_^−^, and not favored on TiO_2_ surface [82]. NO_2_^−^ could be easily oxidized with ^•^OH to NO_3_^−^ (k_NO2− + •OH_ = 6.0 × 10^9^ M^−1^ s^−1^ [83]), thus its concentration remains low.

To understand the reason for the unique behavior of the SMP, we have investigated the effect of wavelength and photocatalyst on the ^•^OH formation rate and charge separation efficiency, using COU and 1,4-BQ.

### 3.4. Transformation of Coumarin—The Comparison of ^•^OH Formation Efficiency

The reaction of COU with ^•^OH (k_COU+•OH_ = 6.9 × 10^9^ M^−1^ s^−1^ [84]) results in highly fluorescent 7-hydroxy-coumarin (7-HC) [85,86]; its formation rate is proportional to the ^•^OH formation rate. The ratio of COU transformation rate and 7-HC formation rate provides further information about the contribution of ^•^OH to the COU transformation [85,87,88].

The COU adsorption was negligible for both photocatalysts (<1.0%), similar to sulfonamides. The effect of the catalyst dose was determined in previous measurements; the r_0_^COU^ reached a maximum value at 1.0 g dm^−3^ in the case of both photocatalysts. In this work, the initial concentration of COU was 1.0 × 10^−4^ M, and the catalyst dosage was 1.0 g dm^−3^.

The transformation of COU was slightly faster for ZnO, especially in the case of LED_398nm_, while the formation rate of 7-HC was significantly higher for TiO_2_, than for ZnO in both cases (Table 2). The maximum concentration of this hydroxylated product is almost twice for TiO_2_ than for ZnO in the case of LED_365nm_ and more than three times higher in the case of LED_398nm_ (Figure 7a,b). The r_0_^COU^/r_0_^7-HC^ ratio for TiO_2_ (0.027 and 0.041 for LED_365nm_ and LED_398nm_, respectively) also exceeds the value determined for ZnO (0.019 and 0.025 for LED_365nm_ and LED_398nm,_ respectively). These prove the higher contribution of ^•^OH to the transformation when TiO_2_ is used, mainly when LED_398nm_ is applied (Table 2). Most probably, for ZnO the transformation of COU via direct charge transfer processes is favorable due to the higher electron mobility of photogenerated charges [89,90].

The apparent quantum yield (Φ^COU^) relates to the COU transformation is higher for TiO_2_ than for ZnO (Table 2); opposite that, the light absorption properties of ZnO are more favorable at 365 nm (Figure 1b). When LED_398nm_ is used, primarily the rutile phase can be excited due to its lower bandgap. Since the rutile content of TiO_2_ is only 15%, the Φ^COU^ is significantly lower at this wavelength than for ZnO (Table 2). Comparing the Φ^COU^ value determined at two different wavelengths was 7 times higher for TiO_2_, but only 3 times higher for ZnO at 365 nm than 398 nm. Similar ratios can be observed for Φ^7-HC^ values. The different wavelengths and photon flux can affect photogenerated charge carriers’ formation and recombination rate [91,92], affecting the quantum efficiency of ^•^OH formation.

The efficiency of both photocatalysts was lower at 398 nm, but despite the nearly 80% reflection and the wide bandgaps (3.2 eV), pretty good activities were measured compared to the irradiation at 365 nm. It is probably due to the presence of rutile for TiO_2_ and the heterojunction between the rutile and anatase phases. The results on COU conversion confirmed that ^•^OH formation is much more efficient for TiO_2_ than for ZnO, and that wavelength also has a significant effect on ^•^OH formation efficiency but does not explain the behavior of SMP.

### 3.5. Transformation of 1,4-BQ—The Comparison of Charge Separation Efficiency

Besides ^•^OH formation efficiency, the possibility of direct charge transfer and efficiency of e_CB_^−^ was also studied and compared. Fónagy et al. demonstrated that 1,4-benzoquinone (1,4-BQ) can be used as a direct e_CB_^−^ scavenger under anoxic atmosphere, and the amount of the formed 1,4-H_2_Q is proportional to that of e_CB_^−^ generated during the excitation of a photocatalyst [93]. Thus, we used the transformation rate of 1,4-benzoquinone (1,4-BQ) and the formation rate of 1,4-H_2_Q to investigate and compare the formation rate of photogenerated e_CB_^−^ The backward reaction is also possible; reaction between 1,4-H_2_Q and h_VB_^+^ (Equation (2)) results in 1,4-BQ [93,94].
1,4-BQ + 2 e_CB_^−^ + 2 H^+^ → 1,4-H_2_Q(1)
1,4-H_2_Q + 2 h_VB_^+^ → 1,4-BQ + 2 H^+^(2)
h_VB_^+^ + OH^−^/H_2_O → ^•^OH(3)
1,4-BQ + ^•^OH → Products(4)
1,4-H_2_Q + ^•^OH → Products(5)

The transformation rate of 1,4-BQ via direct charge transfer in O_2_-free suspension highly exceeds that of the COU transformation, mainly based on reactions initiated by ^•^OH in O_2_-containing suspension (Table 2). The difference between the efficiency of photocatalysts was observed when LED_398nm_ was applied; the 1,4-BQ transformation was faster for ZnO than TiO_2_ (Figure 7d), most probably due to the higher electron mobility of ZnO [89,90]. The transformation was slower for both TiO_2_ and ZnO using 398 nm light due to intense reflection at this wavelength and generating fewer e_CB_^−^ − h_VB_^+^ pairs than 365 nm photons. In both cases, the maximum concentration of 1,4-H_2_Q was higher for TiO_2_ and just slowly transformed. The 1,4-BQ was present (c > 2.3 (±0.2) × 10^−6^ M) during the whole treatment time, clearly indicating the backward reaction via h_VB_^+^ (Equation (2)). In the absence of O_2,_ the reformed 1,4-BQ (Equation (2)) acts as an e_CB_^−^ acceptor (Equation (1)), which opens up the way for ^•^OH formation (Equation (3)). Most probably, the slow decrease of the sum of 1,4-BQ and 1,4-H_2_Q concentrations is caused by ^•^OH initiated transformation (k_1,4-BQ •OH_ = 1.2 × 10^9^ M^−1^ s^−1^ [95] (Equation (4); k_1,4-H2Q + •OH_ = 5.2 × 10^9^ M^−1^ s^−1^ [96] (Equation (5)) (Figure 7d).

Sulfonamides react fast not only with ^•^OH but also e_aq_^−^. Mezyk et al. investigated the kinetics and efficiencies of ^•^OH and e_aq_^−^ based reactions to the transformation of four different sulfa drugs (sulfamethazine, sulfamethizole, sulfamethoxazole, and sulfamerazine) [97]. The rate constants of ^•^OH based oxidation (7.8–8.5 × 10^9^ M^−1^ s^−1^) and degradation efficiencies were similar (changed from 35% to 53%). The rate constants of reduction with the e_aq_^−^ (1.0–2.1 × 10^10^ M^−1^ s^−1^) was even higher and varied within small ranges, but the corresponding degradation efficiency resulted in highly different values from 0.5% to 71%. They proposed that ^•^OH adds predominantly to the sulfanilic acid ring, while reaction with e_aq_^−^ occurs at different reaction sites of the different heterocyclic rings. The higher transformation rate of SMT and SMP, for ZnO than TiO_2_ is probably due to their direct reaction with photogenerated charges. Although the significant contribution of direct charge transfer to the conversion is a possible way to explain the difference of transformation rates observed between TiO_2_ and ZnO, it is difficult to interpret the behavior of SMP in the case of TiO_2_/LED_398nm_ by this way considering that the number of photogenerated charges is much higher for LED_365nm_.

### 3.6. Reaction Mechanism—Effect of Radical Scavenger, Dissolved O_2,_ and the Quality of TiO_2_

Sulfonamides react fast with ^•^OH (k_SMT + •OH_ = 8.3 × 10^9^ M^−1^ s^−1^ [97]); thus, the effect of methanol (MeOH) as ^•^OH-scavenger (k_MeOH + •OH_ = 9.7 × 10^8^ M^−1^ s^−1^ [98]) was investigated. The addition of 2.5 × 10^−3^ M MeOH to 1.0 × 10^−4^ M SMT or SMP scavenges more than 70% of ^•^OH reduced the transformation rates to about half in each case (Figure 8) for TiO_2_/LED_398nm,_ the SMP transformation was decreased to a similar value than TiO_2_/LED_365nm_ or ZnO/LED_365nm_. MeOH can also be used as a h_VB_^+^ scavenger [82,99,100]; therefore, it might prevent the direct oxidation of SMP, but this is not a likely explanation in this case. The direct energy transfer is supposed to be the main reason for the photocatalytic isomerization of trans-caffeic acid in TiO_2_ suspension [26]. The addition of MeOH completely inhibited the transformation in that case; the effect was much larger than expected based on the radical scavenging capacity of MeOH, similar to its effect on SMP transformation for TiO_2_/LED_398nm_.

Both energy transfer and direct charge transfer could increase the conversion rate, change the reaction pathway, and alter the quality of the primary intermediate. The unique behavior of SMP for TiO_2_/LED_398nm_ needed further investigation to clarify since the effect of MeOH was not enough for its proper interpretation, although its high impact indirectly confirmed the role of direct energy transfer.

In O_2_ containing suspension, e_CB_^−^ reacts with molecular O_2_, which is a source of the ROS formation. The further transformation of O_2_^•−^ creates a possibility to the ^•^OH-formation via H_2_O_2_, while in O_2_-free suspension ^•^OH formation is limited to the reaction of H_2_O/OH^−^ with h_VB_^+^. In the O_2_-free suspension, the initial conversion of SMP is slower but still very significant, especially since most organic compounds are not converted at all without O_2_. The shape of the kinetic curves is similar in both O_2_-free and aerated suspension: after 45% and 75% removal of SMP, its further transformation became very slow, almost negligible (Figure 9). In both cases, the main intermediate forms via desulfonation, the concentration of hydroxylated products is negligible. The transformation in O_2_-free suspension via direct charge transfer can take place if SMP could behave as e_CB_^−^ scavenger instead of O_2_, and both oxidation and reduction of SMP can happen similar to the double electron transfer suggested for the formation of singlet oxygen by Nosaka et al. [21]. Thus, two processes are possible: one is the oxidation and reduction of SMP taking place parallel when different parts of the molecule react with e_CB_^−^ and h_VB_^+^ or the direct energy transfer, when excited photocatalysts particles transfer energy to SMP, which is finally transformed. If the reaction with ROS or direct charge transfer process is primarily responsible for transforming an organic compound, its conversion is generally negligible in O_2_-free suspension. However, the rapid desulfonation of SMP in an O_2_-free suspension confirms that the energy transfer is primarily responsible for its conversion. Of course, the contribution of the reactions with the formed ROS in the presence of O_2_ are not negligible.

After 5 and 10 min, there is a “breaking point” on the kinetic curves; after that, the SMP transformation is inhibited, especially in O_2_ containing suspension. The energy transfer can take place via a different mechanism. In the case of “trivial energy transfer”, SMP absorbs the photon emitted by the excited semiconductor. Unlike the trivial mechanism, Förster and Dexter energy transfers are radiationless processes and strongly depend on the distance of the acceptor from the semiconductor. No electron exchange between acceptor and donor occurs when Förster mechanism takes place, while Dexter energy transfer mechanism occurs when simultaneously two electrons move in opposite directions without net charge exchange. The formed SO_3_^2−^/SO_4_^2−^ species and other ions and organic intermediates can adsorb on the TiO_2_ surface [101]. The change of catalysts surface can impede the access to the surface and thus energy transfer via Förster or Dexter mechanism. Moreover, the poisoning of the catalyst surface can also reduce efficiency [102].

The synergistic effect between the two crystal phases resulting in enhanced charge separation and photocatalytic activity was proved by several authors [32,33,103]. Under 398 nm irradiation, mainly the rutile phase excited (Figure 1b), the photogenerated e_CB_^−^ can migrate to the anatase phase, inhibiting the recombination of photogenerated charges [104]. Although heterojunction is controversial, the great activity of P25 is undoubted. For determining whether the unique behavior of the SMP is manifested in the case of anatase or rutile phase or characteristic only to the P25, we examined the conversion of SMP in the presence of pure anatase and rutile phase. But only a slow transformation occurred (Figure 9). The intensive desulfonation was not observed. Mainly hydroxylated products formed indicating a ^•^OH based reaction pathway in the case of anatase and rutile.

From this, it can be concluded that the unique transformation of SMP is only possible for TiO_2_ P25 and is most likely a consequence of the energy transfer, which takes place in this particular case, using P25 photocatalyst and 398 nm radiation. However, the role of direct charge transfer cannot be excluded, and in O_2_-containing suspensions, the ^•^OH-based reaction is likely to contribute to the transformation and mineralization of the products.

### 3.7. Effect of Matrices on the Removal of Sulfonamides

Experiments were performed with SMT and SMP in two water matrices: tapwater with a low organic and high inorganic content, and biologically treated and filtered domestic wastewater (BTWW) with higher organic and inorganic content (Appendix A). For investigation of the inorganic components’ effect, transformation rates were determined in suspensions containing the two most abundant anions, Cl^−^ (120 mg dm^−3^) and HCO_3_^−^ (525 mg dm^−3^). The concentration of the anions was set to the values measured in the biologically treated wastewater (Appendix A).

For SMT, the inhibition effect of matrices was negligible when ZnO photocatalyst was applied, while for TiO_2_ decreased by 25 and 50% using LED_365nm_ and LED_398nm_, respectively (Figure 10a,b). The negative effect of HCO_3_^−^ was more pronounced for ZnO (decrease by 34%) than TiO_2_, while the effect of Cl^−^ is not significant (Figure 10a). A more enhanced inhibition by the matrices was observed in the case of LED_398nm_. For SMP, similar observations can be made, except TiO_2_/LED_398nm_, when both matrices and HCO_3_^−^ inhibited the SMP transformation completely (Figure 10b,d). Even Cl^−^ is inhibited the conversion, while it had no effect in other cases (Figure 10d).

HCO_3_^−^ is a well-known ^•^OH scavenger, but its reaction rate constant is relatively low (k_HCO_3_^−^ + ^•^OH_ = 1.0 × 10^−7^ M^−1^ s^−1^ [98]); therefore, it cannot effectively compete for ^•^OH with the sulfonamides. HCO_3_^−^ also reported as an efficient scavenger of h_VB_^+^, resulting in the formation of carbonate radicals (CO_3_^•−^) [61,105] on the surface of TiO_2_. The formed CO_3_^•−^ is a more selective oxidant than ^•^OH, but sulfonamides, especially SMP, are reported to react with it with a high reaction rate constant (k_SMT + CO_3_^•−^_ = 4.37 × 10^8^ M^−1^ s^−1^, k_SMP + CO_3_^•−^_ = 8.71 × 10^8^ M^−1^ s^−1^ [106]). The negligible effect of HCO_3_^−^ during most of the measurements, and the relatively low effect of the matrices on the transformation rates might be due to the reaction of sulfonamides with formed CO_3_^•−^.

The Cl^−^ does not react with h_VB_^+^, and generally has a negligible effect on photocatalytic activity using TiO_2_ [61]. It may react with ^•^OH with a high reaction rate (k = 4.3 × 10^9^ M^−1^ s^−1^ [107]) to form chlorine radicals, but at neutral and alkaline pH the backward reaction leading to the reformation of ^•^OH is favored [62]. In the case of ZnO, adsorption of Cl^−^ on the surface can promote the separation of photogenerated charges, leading to higher photocatalytic efficiency [108]. Its significantly negative effect can be observed only for SMP, using TiO_2_/LED_398nm_ process. The transformation of SMP using TiO_2_ and LED_398nm_ is most likely caused by the energy transfer—with some contribution of the charge transition—which is especially sensitive to the surface conditions.

## 4. Conclusions

Heterogeneous photocatalysis is generally considered to be based primarily on electron transfer reactions at the surface of the irradiated semiconductor. Besides radical-based reactions and direct charge transfer, attention recently moved towards photocatalytic syntheses, requiring selective transformation. Some examples have been reported in the literature when the transformation is due to the energy transfer between excited semiconductor as energy donor, and results in selective transformation of organic substances.

The present work compares the efficiency of the commercially available photocatalysts, TiO_2_ and ZnO irradiated with 365 nm or 398 nm. Two sulfonamide antibiotics, SMT and SMP, were used to compare the efficiency of the photocatalysts under 398 and 365 nm radiation. The results showed that, besides ^•^OH-based reaction, the direct charge transfer contributes to the transformation even in the case of ZnO. Consequently, the transformation was faster when ZnO was applied; however, TiO_2_ was more efficient in mineralization. The unique behavior, an exceptionally fast transformation of SMP was observed, in the case of TiO_2_/LED_398nm_ process, in contrast to the other cases when 365 nm light was more efficient than 398 nm light. The transformation of SMP was fast in both O_2_ containing and O_2_-free TiO_2_ suspensions and takes place via desulfonation, while in other cases, mainly hydroxylated products form. The effect of reaction parameters confirmed that a quite rarely observed direct energy transfer between the excited state P25 and SMP is likely responsible for this unique behavior; however, the role of direct charge transfer cannot be excluded completely.

Our results have highlighted that “non-conventional” mechanisms can occur in exceptional cases during heterogeneous photocatalysis (even in the case of the well-known photocatalyst, such as TiO_2_), and the effect of wavelength is also worth investigating. The presented results may contribute to further studies on the application of photocatalysts, either in the selective removal of organic pollutants or in the field of organic chemical synthesis.

## Figures and Tables

**Figure 1 materials-15-00049-f001:**
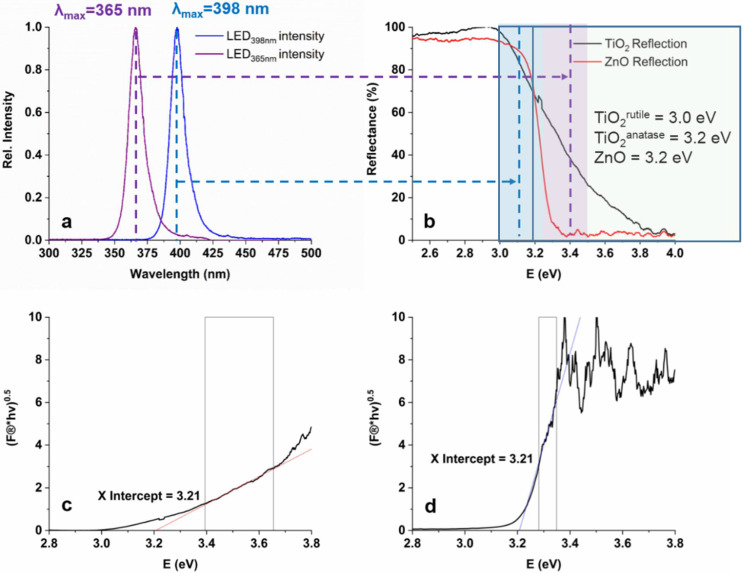
The UV-Vis emission spectra of the LEDs (**a**), the diffuse reflectance spectra and bandgap energies of TiO_2_ and ZnO (**b**), and the Tauc plot originated from DRS spectra for calculation of band gap energies of TiO_2_ (**c**) and ZnO (**d**).

**Figure 2 materials-15-00049-f002:**
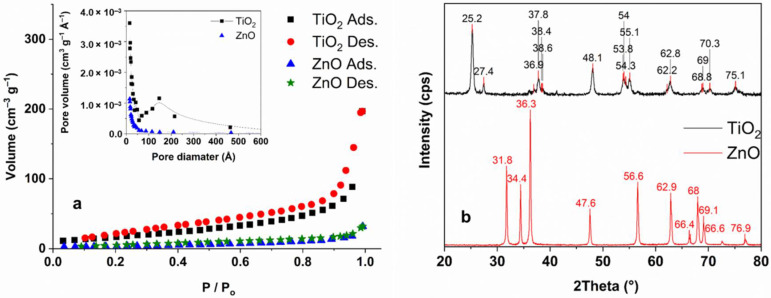
The N_2_ adsorption-desorption isotherms (**a**) with corresponding pore-size distribution according to the BJH model (inset), and the XRD patterns of TiO_2_ and ZnO (**b**).

**Figure 3 materials-15-00049-f003:**
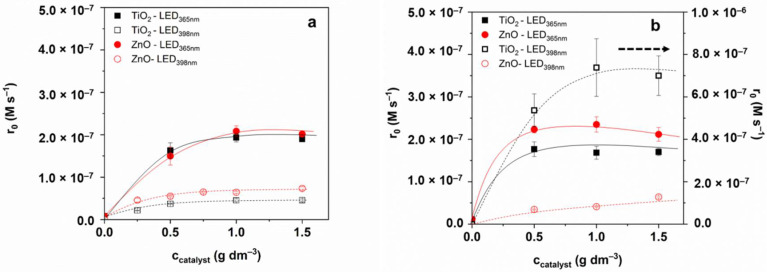
The effect of photocatalyst concentration on the initial transformation rate of SMT (**a**) and SMP (**b**).

**Figure 4 materials-15-00049-f004:**
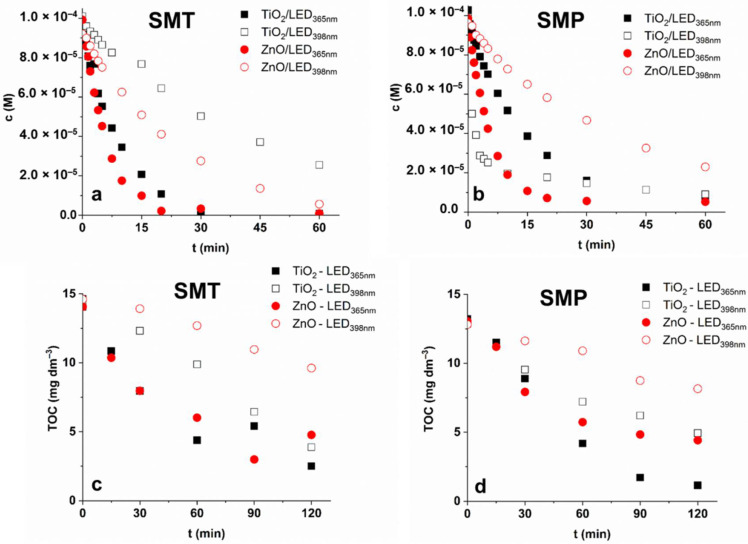
The concentration and TOC content of SMT (**a**,**c**) and SMP (**b**,**d**) during treatments.

**Figure 5 materials-15-00049-f005:**
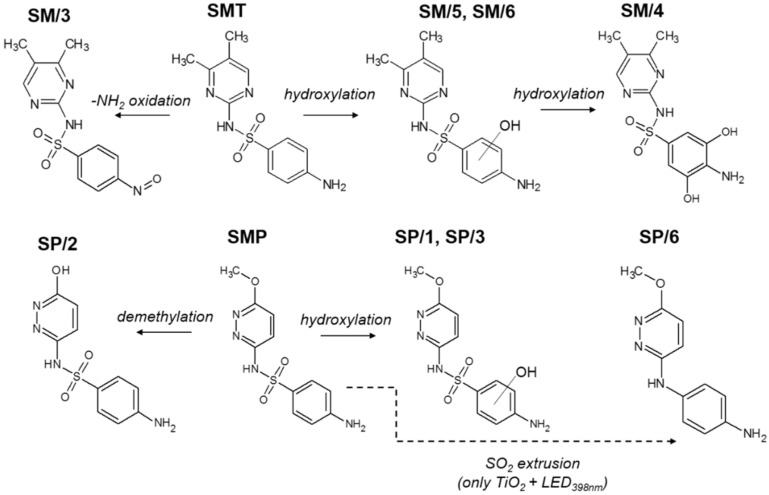
The proposed primary stable products of SMT and SMP transformation during heterogeneous photocatalysis.

**Figure 6 materials-15-00049-f006:**
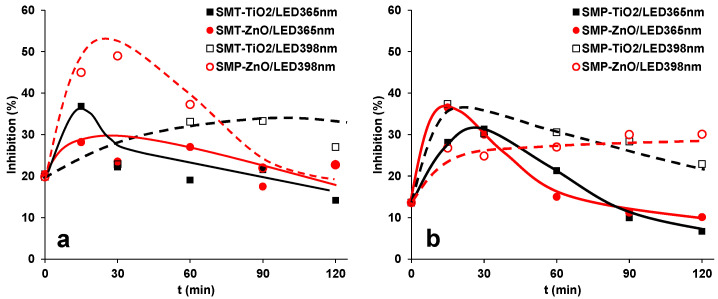
Change of ecotoxicity of the SMT (**a**) and SMP (**b**) solutions as a function of treatment time.

**Figure 7 materials-15-00049-f007:**
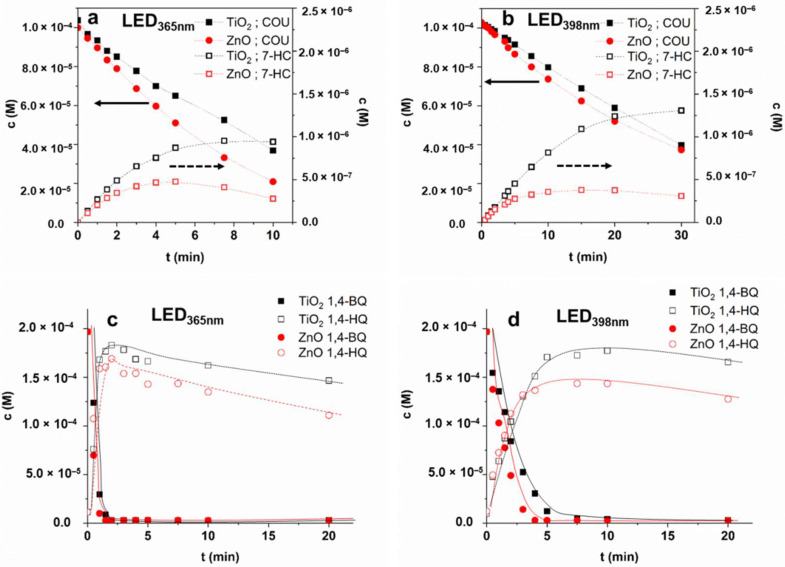
The concentration of COU and 7-HC (**a**,**b**) in aerated suspensions and the concentration of 1,4-BQ and 1,4-H_2_Q (**c**,**d**) in oxygen-free suspensions ((**a**,**c**): LED_365nm_; (**b**,**d**): LED_398nm_).

**Figure 8 materials-15-00049-f008:**
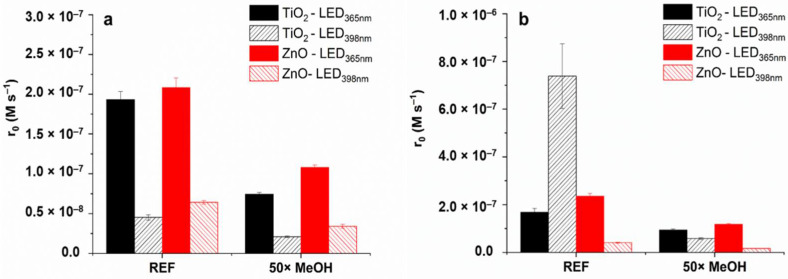
The effect of 2.5 × 10^−3^ M MeOH on the initial transformation rate of SMT (**a**) and SMP (**b**).

**Figure 9 materials-15-00049-f009:**
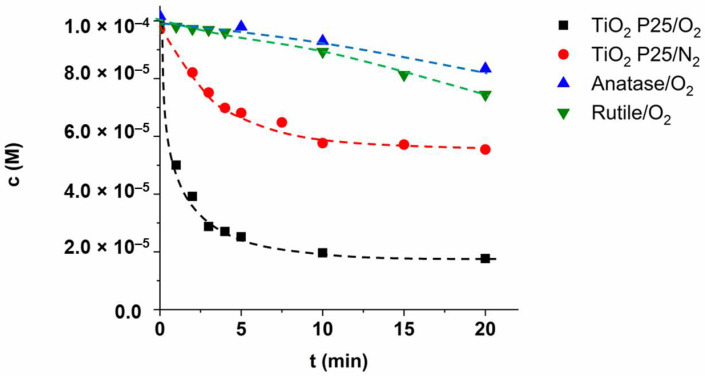
The concentration of SMP in aerated and O_2_-free suspensions using different TiO_2_ photocatalysts and LED_398nm_.

**Figure 10 materials-15-00049-f010:**
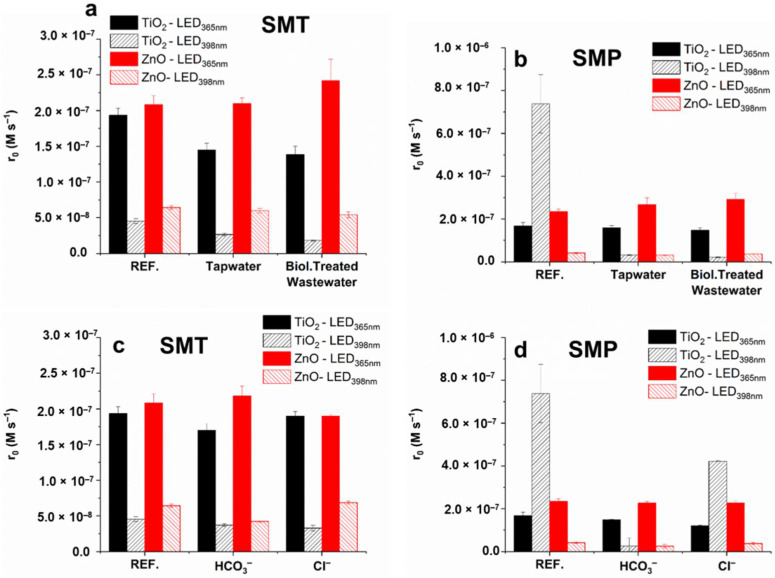
The initial reaction rates of SMT (**a**,**c**) and SMP (**b**,**d**) determined in purified water (REF), real water matrices (tapwater and biologically treated domestic wastewater), and in the presence of Cl^−^ (120 mg dm^−3^) and HCO_3_^−^ (525 mg dm^−3^).

**Table 1 materials-15-00049-t001:** The initial transformation rates of SMT and SMP and the apparent quantum efficiencies (Φ) of the related processes.

	TiO_2_				ZnO			
	SMT		SMP		SMT		SMP	
	r_0_^SMT^ (mol dm^−3^ s^−1^)	Φ^SMT^	r_0_^SMP^ (mol dm^−3^ s^−1^)	Φ^SMP^	r_0_ ^SMT^ (mol dm^−3^ s^−1^)	Φ^SMT^	r_0_^SMP^ (mol dm^−3^ s^−1^)	Φ^SMP^
LED_365nm_	1.93 × 10^−7^	7.0 × 10^−3^	1.68 × 10^−7^	6.1 × 10^−3^	2.08 × 10^−7^	7.6 × 10^−3^	2.35 × 10^−7^	8.5 × 10^−3^
LED_398nm_	4.53 × 10^−8^	9.7 × 10^−4^	7.38 × 10^−7^	1.6 × 10^−2^	6.43 × 10^−8^	1.4 × 10^−3^	4.12 × 10^−8^	8.8 × 10^−4^

**Table 2 materials-15-00049-t002:** The initial transformation rates of target substances (r_0_^COU^ and r_0_^1,4-BQ^), the initial formation rate of their primary products (r_0_^7-HC^ and r_0_^1,4-H2Q^), and the apparent quantum efficiency (Φ) of the related processes.

	TiO_2_				ZnO			
	COU → 7-HC							
	r_0_^COU^ (mol dm^−3^ s^−1^)	Φ^COU^	r_0_^7-HC^ (mol dm^−3^ s^−1^)	Φ^7-HC^	r_0_^COU^ (mol dm^−3^ s^−1^)	Φ^COU^	r_0_^7-HC^ (mol dm^−3^ s^−1^)	Φ^7-HC^
LED_365nm_	1.53 × 10^−7^	5.5 × 10^−3^	4.1 × 10^−9^	1.5 × 10^−4^	1.54 × 10^−7^	3.2 × 10^−3^	2.88 × 10^−9^	0.6 × 10^−4^
LED_398nm_	3.80 × 10^−8^	0.8 × 10^−3^	1.5 × 10^−9^	0.3 × 10^−4^	5.18 × 10^−8^	1.1 × 10^−3^	1.29 × 10^−9^	0.3 × 10^−4^
	1,4-BQ → 1,4-H_2_Q							
	r_0_^BQ^ (mol dm^−3^ s^−1^)	Φ^1,4-BQ^	r_0_^1,4-H^_2_^Q^ (mol dm^−3^ s^−1^)	Φ^1,4-H^_2_^Q^	r_0_^BQ^ (mol dm^−3^ s^−1^)	Φ^1,4-BQ^	r_0_^1,4-H^_2_^Q^ (mol dm^−3^ s^−1^)	Φ^1,4-H^_2_^Q^
LED_365nm_	2.78 × 10^−6^	1.0 × 10^−1^	2.61 × 10^−6^	9.5 × 10^−2^	3.12 × 10^−6^	1.1 × 10^−1^	2.47 × 10^−6^	8.9 × 10^−2^
LED_398nm_	8.88 × 10^−7^	1.9 × 10^−2^	7.61 × 10^−7^	1.5 × 10^−2^	1.19 × 10^−6^	2.6 × 10^−2^	8.07 × 10^−7^	1.7 × 10^−2^

## Data Availability

The data is included in the article or Appendix A.

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
