# Peer review of "Wavelength Dependence of the Transformation Mechanism of Sulfonamides Using Different LED Light Sources and TiO_2_ and ZnO Photocatalysts"

_materials, 2021, doi:10.3390/ma15010049_

Round 1
Reviewer 1 Report
File attached..

Author Response
Dear Reviewer,
At first, let me thank you for reviewing our manuscript and enhancing the quality of our paper with your valuable remarks and suggestions.
The attached file contains our answers.
"Please see the attachment."

Reviewer 2 Report
I have received and reviewed the paper entitled “Wavelength dependence of the transformation mechanism of sulfonamides using different LED light sources and TiO2 and ZnO photocatalysts”. In this paper, the authors address the photocatalytic degradation of two antibiotics; sulfamethazine and sulfamethoxypyridazine, using two well establish photocatalysts, ZnO and TiO2. Their study mainly focuses on how the physicochemical properties of the materials are affected by the LED light sources, in terms of removal and mineralization efficiency. Moreover, they present a comprehensive analysis of the degradation mechanism and the by-products formation, so as their alteration under different conditions. In my opinion, this manuscript is relevant with the Journal’s topics and it is worth noticing that the authors i) examined in depth and explained their findings, ii) they propose a plausible degradation mechanism and iii) they give insights for further investigation in future.
In general, most of the following comments/suggestions are related with wording and text editing, in order to improve the manuscript and help the reader. Thus, a few more essential comments are also included.
Comments
* The only arising concern is related with the last sentences in the Abstract: In section “3.6 Reaction mechanism- …” are presented very interesting results, which could act as an indirect proof of energy transfer and how it could possibly participate in the degradation mechanism (in any case, these phenomena are very difficult to be distinguished directly). So, it could better to change the last sentence to: “…confirmed that a quite rarely observed energy transfer between the excited state P25 and SMP might be responsible for this unique behavior. In our opinion, these results highlight that “non-conventional" mechanisms could occur even in the case of the well-known TiO2 photocatalyst and the effect of wavelength is also worth investigating.”.
* Abstract/page1/line15: for “the” removal
* Abstract/p.1/l.15: mention “benzoquinone in de-oxygenated suspensions”
* Introduction/p.3/l.97: “In addition, extending…”
* Introduction/p.3/l.106: effect of absorbing light of different wavelength
* Introduction/p.3/l.113: “usually significantly reduced efficiency”? Is “usually” needed?
* Introduction/p.3/l.118: The starting sentence has a positive effect, not hindering. Maybe in this paragraph you should compare/mention the positive/negative effects of the matrix. Minor rephrasing is needed in this paragraph, in order to be more clear.
* Materials/p.3/l.144: The UV-A photoreactor has a cooling system? If yes, mention
* Results/p.5/l.228: Not, only here, but everywhere through the manuscript the plots are mentioned as Fig.1., not Fig. 1a. Please correct all the corresponding parts through the text (the labels already exist in the figures.).
* Results/p.6/l.246: Delete the”)” after [27].
* Results/p.7/l.254: Rephrase to “…the adsorption of both SMT and SMP was negligible (<2%) for both catalysts.”
* Results/p.7/Figure 3b.: There are two y axis units. I suppose the one of them should be deleted
* Results/p.8/Table 1: This table should be spitted in two: i) One table for SMT and SMP, which should be moved in S.I. as long as there is no comment in the manuscript (except if a comment will be added) and ii) one for COU and BQ, which should be moved to section "3.4. Transformation of coumarin –“
* Results/p.9/l.272: In this sentence you could mention that this examination lasted 60 minutes . (In this way, it will be also clearer why the TOC experiments were performed for longer period)
* Results/p.9/l.285: “than for LED365nm” is needed in the sentence?
* Results/p.9/l.298: the (3DOM) should be deleted?
* Results/p.10/Figure 6: The labels “a” and “b” are missing in the two plots
* Results/p.13/l.415: The H2Q by-product name is not given
* Results/p.14/l.458: Herein in you should mention that the ro are compared with the values without MeOH, which are presented in the Table 1 (if the table is not transferred to SI)
* Results/p.14/l.461: The “365nm” subscript is missing
* Results/p.14/l.487: Practically, the charge transfer mechanism is described again in that sentence. Delete it and write about the alternative “energy transfer” only.
* Results/p.14/l.489: “..particles transform… finally transformed”. I can’t understand if it’s ok or the word “transform” was used twice by mistake.
* Results/p.15/l.509: What you mean by physical mixture? Two different materials, without heterojunction cannot approximate the polycrystalline P25 catalyst.
* Results/p.15/caption of Figure9: the key parameter here is the aerated or de-oxygenated conditions, not the treatment time. Replace
Author Response

(The authors gave the same response as above.)

Round 2
Reviewer 1 Report
Sufficiently improved.